# ReSWD: ReSTIR'd, not shaken.
# Combining Reservoir Sampling and Sliced Wasserstein Distance for Variance Reduction

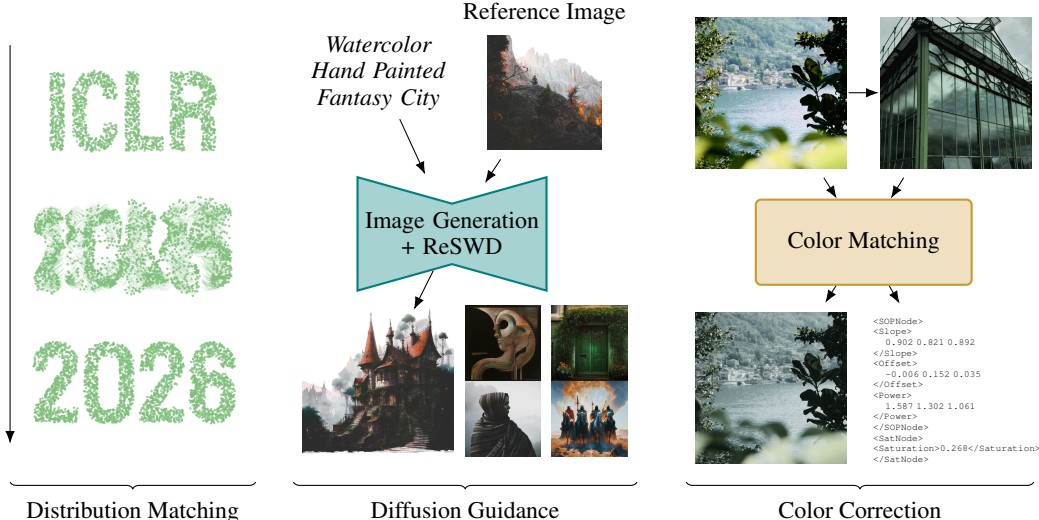

Figure 1: **Applications.** SWD can be used in various applications and with ReSWD we enhance their performances. Here, we show diffusion guidance, color corrections and general distribution matching.

## Abstract

Distribution matching is central to many vision and graphics tasks, where the widely used Wasserstein distance is too costly to compute for high-dimensional distributions. The *Sliced Wasserstein Distance* (SWD) offers a scalable alternative, yet its Monte Carlo estimator suffers from high variance, resulting in noisy gradients and slow convergence. We introduce *Reservoir SWD* (ReSWD), which integrates Weighted Reservoir Sampling into SWD to adaptively retain informative projection directions in optimization steps, resulting in stable gradients while remaining unbiased. Experiments on synthetic benchmarks and real-world tasks such as color correction and diffusion guidance show that ReSWD consistently outperforms standard SWD and other variance reduction baselines.

## 1 Introduction

Distribution matching is a central problem in computer vision and graphics: from classical tasks such as histogram matching to more nuanced applications such as color grading (Pitie et al., 2005; Rabin et al., 2012; Bonneel et al., 2015; He et al., 2024), texture alignment (Elnekave & Weiss, 2022; Heitz et al., 2021), and the guidance of generative models Lobashev et al. (2025). Among the available metrics, the Wasserstein distance has emerged as particularly powerful due to its ability to capture shifts between distributions. However, the Wasserstein distance suffers from the curse of dimensionality and has a practical convergence rate scaled with the d-dimensional distribution $\mathcal{O}\left(n^{-\frac{1}{d}}\right)$, with $n$ data points. This limits its direct use in many iterative optimization settings.

The *Sliced Wasserstein Distance* (SWD) (Pitie et al., 2005; Bonneel et al., 2015; Heitz et al., 2021) offers a practical alternative by projecting distributions in random directions and averaging their 1-D Wasserstein costs. This reduces the computational burden to a sequence of 1-D sorting problems but introduces a new challenge: the expectation over projection directions is typically approximated via Monte Carlo (MC) sampling, which suffers from high variance. Increasing the directions would result in the true expected value, but the complexity of SWD scales with the number of random directions $M$ as $\mathcal{O}(Mn \log n)$. Hence, the curse of dimensionality is introduced again indirectly with the number of projections, which should increase with the dimensionality of the data. In optimization scenarios, where such SWD metrics are used as loss functions to optimize neural networks, this variance directly results in noisy gradients and slower convergence.

A variety of variance reduction techniques have been explored for MC estimators, but most remain underutilized in distribution matching objectives. In this work, we draw inspiration from recent advances in rendering, particularly the ReSTIR resampling framework (Bitterli et al., 2020), and adapt *Weighted Reservoir Sampling* (WRS) (Efraimidis & Spirakis, 2006; Chao, 1982) to the SWD setting. The resulting estimator, which we term *Reservoir SWD* (ReSWD), continuously reuses and reweighs the most informative projection directions throughout the optimization. Intuitively, ReSWD preferentially retains those directions where the two distributions are most dissimilar, thus concentrating computational effort on projections that produce stronger and more stable gradients. This simple but effective modification significantly reduces stochastic variance while preserving unbiasedness due to the weighting of the WRS, leading to faster and more robust optimization.

We demonstrate the advantages of ReSWD on two real-world distribution matching problems of color correction and color diffusion guidance, as shown in Fig. 1, as well as synthetic general distribution matching problems, showing clear improvements over the standard SWD and existing variance reduction baselines.

## 2 RELATED WORK

**Distribution Matching and Optimal Transport.** Distribution matching is the general task of aligning two distributions. A particularly effective way of achieving this is with *Optimal Transport* (OT) by minimizing the cost of moving the probability mass, which is often performed using Wasserstein distances (Villani et al., 2008; Peyré & Cuturi, 2019). Entropic regularization accelerates OT through Sinkhorn iterations (Cuturi, 2013), leading to scalable solvers and Sinkhorn divergences that interpolate between OT and kernel *Maximum Mean Discrepancies* (MMD) (Feydy et al., 2019). OT has been widely applied to textures, color transfer, and barycenters (Rabin et al., 2012; Bonneel et al., 2015; Pitie et al., 2005).

To address high-dimensional costs, *Sliced Wasserstein* (SW) replaces couplings with averages of 1-D transports. Early works introduced SW barycenters and Radon/Projected variants that are fast and differentiable (Bonneel et al., 2015; Rabin et al., 2012). Extensions include Max-SW, which selects the most contributing directions instead of averaging over them (Deshpande et al., 2019), and several other techniques to improve efficiency and optimization performance (Kolouri et al., 2019; Nguyen et al., 2021; 2023; Nguyen & Ho, 2023; Nguyen et al., 2024b). SW has also been studied through geometry and flows: gradient flows as generative dynamics with long-time analysis (Cozzi & Santambrogio, 2024; Vauthier et al., 2025), intrinsic geometry of SW space (Park & Slepčev, 2025), extensions to Cartan–Hadamard manifolds (Bonet et al., 2025), and stereographic spherical SW for curved domains (Tran et al., 2024). Recent work further reduces variance via quasi-Monte Carlo and control variates (Nguyen et al., 2024a; Nguyen & Ho, 2024). In practice, SW is widely used as a loss in graphics, vision, and generative modeling *e.g.*, for textures, patch statistics, and color transfer due to stable gradients and favorable sample complexity (Heitz et al., 2021; Elnekave & Weiss, 2022; Pitie et al., 2005; Wu et al., 2024).

Our work ties into the work selecting the most contributing directions similar to Max-SW but achieves this by building a reservoir of high contributing directions during the training. This improves efficiency, as we keep more directions in optimization. It also ties in with the work on variance reduction while remaining unbiased, as our technique is inspired by variance reduction techniques in real-time path tracing (Bitterli et al., 2020).

**Color Transfer and Correction.** Classical color transfer methods framed recoloring as histogram or distribution matching across images, often using statistical metrics or optimal transport (Pitie et al., 2005; Rabin et al., 2012; Bonneel et al., 2015). These techniques established the foundation for perceptual color differences in imaging. More recently, multiscale OT and Wasserstein-based metrics have been proposed as perceptually faithful color difference measures (He et al., 2024).

With deep learning, style transfer methods enabled more expressive color and appearance manipulation. CNN-based approaches introduced artistic and photorealistic stylization (Gatys et al., 2016; Luan et al., 2017; Huang & Belongie, 2017; Li et al., 2017; 2018; Yoo et al., 2019; An et al., 2020; Chiu & Gurari, 2022; Hong et al., 2021), with adaptive instance normalization and feature transforms widely adopted to control color and tone. These methods evolved from hand-crafted global mappings to neural architectures capable of spatially aware and semantically coherent color control.

With the advance of text-guided diffusion models (Rombach et al., 2022b;a), additional mechanisms to control the style such as LoRA (Hu et al., 2021), IP-Adapter (Zhang et al., 2023b), ControlNets (Zhang et al., 2023a) are introduced. A recent work by Lobashev *et al*. (Lobashev et al., 2025) also demonstrated that traditional constraints based on OT can be used during generation to enforce a color distribution.

Our work can be used to enhance traditional color matching works with improved efficiency and to enable more novel color-guided diffusion tasks.

## 3 METHOD

In our setting, we only ever compare empirical *discrete* distributions, that is, two sets of samples $X = \{x_1, \dots, x_N\} \subset \mathbb{R}^d$ and $Y = \{y_1, \dots, y_N\} \subset \mathbb{R}^d$, drawn from the underlying distributions $\mathcal{X}$ and $\mathcal{Y}$.

### 3.1 PRELIMINARIES

**Wasserstein Distance.** The Wasserstein $p$-distance measures differences between two empirical distributions. In the 1D case relevant to our setting, given two sets of samples $X$ and $Y$, the Wasserstein distance can be computed simply by sorting both sets and comparing the corresponding order statistics:

$$\mathrm{W}_p(X, Y) = \left( \frac{1}{n} \sum_{i=1}^{n} |x_i - y_i|^p \right)^{1/p}, \tag{1}$$

This efficient formulation has complexity $\mathcal{O}(n \log n)$ due to sorting, in contrast to the cubic complexity of the general $d$-dimensional case (Rabin et al., 2012; Bonneel et al., 2015; Pitie et al., 2005).

**Sliced Wasserstein Distance (SWD).** The Sliced Wasserstein Distance proposes to stochastically approximate the true Wasserstein distance in multi-dimensional distributions. It is defined as (Pitie et al., 2005; Heitz et al., 2021; Elnekave & Weiss, 2022):

$$\mathrm{S}_p(X, Y) = \mathbb{E}_{\theta \sim U(S^{d-1})} \Big[ \mathrm{W}_p(\pi_\theta X, \pi_\theta Y) \Big], \tag{2}$$

where $\pi_\theta$ indicates the projection onto the unit-normal vector $\theta$ (uniformly sampled from $S^{d-1}$, the $d$-dimensional unit sphere $\mathbb{R}^d$).

In practice, the expectation is often approximated via Monte Carlo (MC) integration:

$$\mathrm{S}_p(X, Y) \approx w_i \sum_{i=1}^{L} \mathrm{W}_p(\pi_{\theta_i} X, \pi_{\theta_i} Y). \tag{3}$$

with random uniformly sampled directions $\theta_i$ and $w_i$ being the sampling weights, *i.e.* $w_i = \frac{1}{L}$ with uniform direction sampling. The resulting complexity is therefore $\mathcal{O}(Ln \log n)$. The SWD is an unbiased estimate of the true Wasserstein distance (Pitie et al., 2005; Heitz et al., 2021). With modern frameworks such as PyTorch or Tensorflow, the entire metric can be trivially implemented fully differentiable, which enables usage in optimization settings. However, due to the MC integration, SWD can lead to noisy gradients, which we aim to solve with ReSWD.

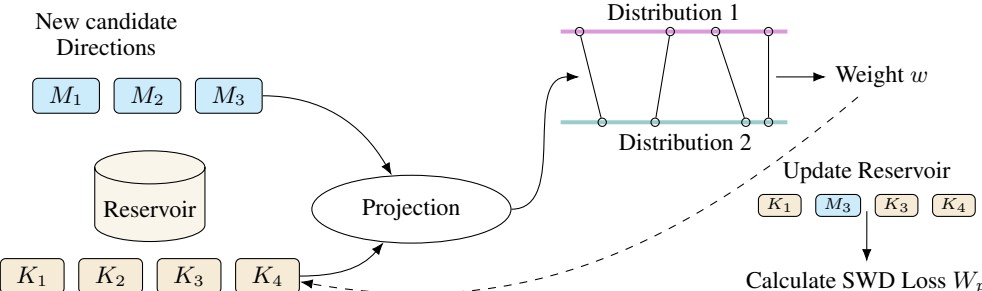

Figure 2: **Overview.** During the optimization we keep a reservoir of highly influential directions. We use the Sliced Wasserstein Distance (SWD) as a proxy metric for the reservoir update and the final optimization loss. Note that only directions in the reservoir influence the optimization to remain unbiased.

### 3.2 RESERVOIR SLICED WASSERSTEIN DISTANCE (RESWD)

Although SWD is highly efficient in the calculation at high dimensions, it still suffers from high variance due to the MC integration from random directions. Especially, when SWD is used as a loss during an optimization, this leads to noisy gradients. In computer graphics, variance is often also an issue, especially in real-time path tracing where MC integration is also usually employed. Inspired by ReSTIR's *resampled-importance-sampling* (Bitterli et al., 2020), we incorporate the Weighted Reservoir Sampling (WRS) mechanism into SWD. This allows the reuse of information between optimization steps, and hence faster convergence overall with minimal performance penalties.

Weighted Reservoir Sampling is a family of algorithms that draw a fixed subset from a weighted stream of candidates, *i.e.* directions, in a single pass (Chao, 1982; Efraimidis & Spirakis, 2006). Suppose that we wish to maintain a reservoir of $K$ candidates while processing a candidate pool of $K + M$ items, where $M$ denotes the number of newly drawn candidates. Each candidate $\theta_j$ is associated with a non-negative weight $w_j$, and the goal is to select exactly $K$ survivors such that the marginal inclusion probability of each element is proportional to $w_j$. This is achieved by assigning to every candidate a random key (Efraimidis & Spirakis, 2006)

$$k_j = u_j^{1/w_j}, \qquad u_j \sim \mathcal{U}(0,1), \tag{4}$$

retaining the $K$ elements with the smallest keys. The resulting selection is unbiased: the expected inclusion probability of each element is proportional to its weight, while the reservoir size remains fixed (Efraimidis & Spirakis, 2006). In our context, WRS allows efficient reallocation of computational effort toward projections with higher contribution to the optimization loss, while preserving Monte Carlo unbiasedness.

In Fig. 2, we present an overview of our proposed method. Let $\mathcal{P}$ be a pool of $K + M$ 1-D projection directions $\theta \subset S^{d-1}$. For every $\theta \in \mathcal{P}$ we compute the 1-D $p$-*power* Wasserstein cost

$$D(\theta) = W_p\big(\pi_\theta\mu, \pi_\theta\nu\big) \tag{5}$$

Then each distance calculation performs the following steps.

**Step 0: Time-decay reweighting.** As SWD is most often used in optimization scenarios, we introduce $\tau$ as the time decay constant to place more emphasis on newer random directions to account for the shifting optimization field. Before drawing new candidates, we *age* the stored reservoir weights:

$$\tilde{w}_i \leftarrow w_i \exp\big[-(t - t_i)/\tau\big], \qquad \tilde{k}_i \leftarrow k_i \exp\big[-(t - t_i)/\tau\big], \tag{6}$$

where $t_i$ is the step when $\theta_i$ entered the reservoir. This exponential decay (enabled when $\tau > 0$) steadily forgets stale projections so that the sampler adapts to *nonstationary* optimization trajectories.

**Step 1: Reservoir construction.** At optimization step $t$ we maintain a persistent reservoir $\mathcal{R}_{t-1}$ of $K$ directions from the previous optimization step $t - 1$ and draw $M$ new directions $\mathcal{N}_t$. The candidate set is $\mathcal{P}_t = \mathcal{R}_{t-1} \cup \mathcal{N}_t$.

**Step 2: Weighted reservoir sampling.** Following Efraimidis & Spirakis (Efraimidis & Spirakis, 2006) we assign each $\theta \in \mathcal{P}_t$ a key $k(\theta) = u^{1/D(\theta)}$, $u \sim \mathcal{U}(0,1)$ and keep the $K$ largest keys. This selects each $\theta$ with probability $q(\theta) = \frac{D(\theta)}{\sum_{\theta' \in \mathcal{P}_t} D(\theta')}$, *i.e.* proportionally to its contribution to the loss.

**Step 3: Self-normalized weights.** For the survivors $\{\theta_i\}_{i=1}^K$ we calculate the loss with weights as:

$$\widehat{S}_p(\mu, \nu) = \sum_{i=1}^K \underbrace{\frac{1/q(\theta_i)}{\sum_j 1/q(\theta_j)}}_{w_i} D(\theta_i), \tag{7}$$

This loss can then be used in optimizations with a detached gradient calculation for importance weights $w_i$. This remains an *unbiased* MC estimate of $\mathbb{E}_\theta[W_p]$, while focusing the computation on projections that highlight larger differences.

**Step 4: ESS-based reservoir reset.** We monitor the effective sample size (Bitterli et al., 2020) $\text{ESS} = (\sum_i w_i)^2 / \sum_i w_i^2$ and flush the reservoir whenever $\text{ESS} < \alpha K$ ($\alpha = 0.5$ in all experiments), preventing weight collapse.

**Handling unequal sample counts.** If the sample sizes of $X$ and $Y$ are not equal, we repeat the shorter projection so that both 1-D arrays have the same length before sorting. Repetitions are picked uniformly with replacement, following Elnekave & Weiss (2022).

**Complexity.** The dominant cost is sorting $n$ scalars for each of $(K + M)$ directions: $\mathcal{O}\big((K + M)\, n \log n\big)$. Key generation, ESS evaluation, and gradient accumulation are $\mathcal{O}(K + M + d)$.

**Algorithmic summary.** The overview of the algorithm is shown in Algorithm 1.

---

**Algorithm 1** ReSWD estimator (per optimisation step)

---

**Require:** batches $\{x_n\}_{n=1}^{N_\mu}$, $\{y_m\}_{m=1}^{N_\nu}$, reservoir $\mathcal{R}_{t-1}$, hyper-params $K, M, p, \alpha, \tau$
1: **for** $(\theta_i, w_i, k_i, t_i) \in \mathcal{R}_{t-1}$ **do**                                   ▷ Step 0: Time-decay
2:     $w_i \leftarrow w_i \exp\big[-(t - t_i)/\tau\big]$
3:     $k_i \leftarrow k_i \exp\big[-(t - t_i)/\tau\big]$
4: **end for**
5: $\mathcal{N}_t \leftarrow \text{DRAWDIRECTIONS}(M, d)$                                   ▷ Step 1: Reservoir Construction
6: $\hat{x}, \hat{y} \leftarrow \text{DUPLICATESAMPLES}(x_n, y_m)$                         ▷ See "Handling unequal sample counts"
7: **for** $\theta \in \mathcal{N}_t$ **do**                                                        ▷ Step 2: Reservoir Sampling
8:     $D(\theta) \leftarrow W_p(\pi_\theta \hat{x}, \pi_\theta \hat{y})$
9:     $k_t(\theta) \leftarrow u^{1/D(\theta)}$, $u \sim \mathcal{U}(0,1)$
10: **end for**
11: $k(\theta) \leftarrow R_{t-1}[k] \cup k_t(\theta)$                                          ▷ Join reservoir with new keys
12: $\mathcal{R}_t \leftarrow \mathcal{R}_{t-1}[\theta] \cup \mathcal{N}_t$                     ▷ Join reservoir with new directions
13: $\mathcal{R}_t \leftarrow K$ directions with largest $k(\theta)$
14: compute $q(\theta)$, weights $w_i$, and $\widehat{W}_p$ via Eq. (7)                      ▷ Step 3: Self normalizing weights
15: **if** ESS $< \alpha K$ **then return** $\widehat{W}_p$, $\varnothing$                      ▷ Step 4: ESS-Reset
16: **else return** $\widehat{W}_p$, $\mathcal{R}_t$
17: **end if**

---

### 3.3 APPLICATIONS

We showcase our SWD modifications on two real-world applications.

**Color Correction.** Often in movie production two shots have to be matched in terms of their color appearance. We tackle this approach by matching a source image using a Color Decision List (CDL) (Pines & Reisner, 2009) with a reference image. The source image $x$ is first passed through a differentiable CDL implementation, applying the *slope $s$*, *offset $o$*, *power $p$* and $\lambda$ *saturation* adjustments according to $x' = \text{S}\big((s \times x + o)^p; \lambda\big)$. The saturation is defined as $\text{S}(x; \lambda) = \text{L}(x) + \lambda\big(x - \text{L}(x)\big)$ and $\text{L}(x) = 0.2126 x_r + 0.7152 x_g + 0.0722 x_b$, where $r, g, b$ define the color channels. The source and reference are then converted to the CIELAB space. We

found that this space improves the color accuracy, which is consistent with a recent paper proposing a perceptual color metric (He et al., 2024). As we perform the matching on a pixel level, we find that lower resolutions result in faster matching speeds with similar performance. Thus, we leverage a resolution of 128 in the maximum dimension. Within 150 steps, the CDL parameters are optimized and can be applied to the full-res image or even a video clip.

**Diffusion Guidance.** Lobashev et al. (2025) proposed to incorporate SWD guidance in diffusion models to shift the final generation towards a certain color distribution. The main concept is to optimize a small offset on top of the current latents with $i$ SWD steps in each diffusion step. Here, we use the predicted $x_0$ in each step and decode it using the VAE decoder. This prediction is then matched with the reference image. We apply this approach to the more recent flow matching model SD3.5 (Stability AI, 2025). This required several modifications. Lobashev et al. (2025) calculated the full gradient from the input, through the VAE and the U-net. With the increased complexity of larger transformer models, we opted to employ a gradient stop after the backbone similar to SDS (Poole et al., 2022). Hence, we only backpropagate through the VAE decoder. Similarly to our color-correction approach, we also opted to perform the matching in the CIELAB space. In addition, we replace the simple gradient descent of Lobashev et al. (2025) with an Adam optimizer. With these changes, we use a learning rate of $3e-3$ and a total of 6 steps for $95\%$ of the total denoising steps. We do not reset the reservoir after each denoising step, but twice during the generation, as we only perform 6 SWD steps, which would not suffice to build a reservoir. This general method can then be applied to medium, large, and large-turbo SD3.5 models using the recommended CFG and step counts.

## 4 RESULTS

We perform synthetic as well as real-world tests with ReSWD and study the influence of our main hyperparameter, the number of fresh candidates in each optimization step.

**General Distribution Matching.** For a general test of our proposed method we create 1000 $d = 3$ distribution pairs (normal, uniform, bimodal normal) and align them based on 1024 samples in 300 steps. All methods leverage 64 projections. In Table 1, our method clearly outperforms previous SWD techniques with a slight additional performance cost. To evaluate the optimization behavior of our method, we measure the gradient signal-to-noise ratio (SNR) by taking the mean gradient as the signal and the standard deviation as the noise. A higher value here indicates that the method provides more stable gradients. Additionally, we calculate the mean $W_1$ true Wasserstein score for each dimension in each step. This allows us to plot the convergence behavior of our method in Fig. 3. Here, it is evident that pure SWD and our ReSTIR-based modification have a similar trajectory, but building the reservoir initially results in slightly slower trajectory, which produces better results after roughly 140 steps. This also allows us to investigate the correlation between our proposed loss and the true Wasserstein distance. We also investigate the combination of ReSWD with QMC in Table 1 and observe an additional performance improvement with minimal additional overhead. In Fig. 4, our method achieves a high correlation with the true loss, indicating the unbiased nature of our method.

**Color Matching.** To evaluate the color matching based on our differentiable color correction pipeline, we created a dataset of 10 scenes with two different illumination settings each. For each setting, we also take a photo using a calibration color chart. We match the illumination pairs and compute the following metrics from the color checker data extracted from the additional image pairs: Color data PSNR and RMSE (between adjusted and ground truth color checker images), transform error, i.e. deviation from an identity transform between target and adjusted colors as RMSE, as well as an adapted CTQM metric (Panetta et al., 2016) describing the overall color transfer quality. We select Reinhard et al. (2001) and Nguyen et al. (2014) as conventional baselines, Yoo et al. (2019), Ho & Zhou (2021) and Larchenko et al. (2025) as neural network based comparisons, as well as our proposed method but without the addition of ReSWD. Table 3 shows that our approach achieves the lowest transform error in a competitive runtime. In Fig. 5, we present the visual comparison between the methods. It is also evident that our method outperforms the existing baseline in qualitative results as well. Additionally, it is worth pointing out that our results using a parametric model also offer real-world advantages because of better pipeline integration and further manual edits being possible.

Table 1: **Comparison on 1D distribution matching.** Mean-$W_1$ over 1000 distribution matches for various methods alongside the respective running time. Here, we can see that ours provides the best performance with a comparatively low run-time cost.

| Method | Mean-$W_1$ $[10^3]$ ↓ | Grad SNR ↑ | Time per step [ms] ↓ |
|---|---|---|---|
| SWD | 0.733 | 0.215 | **1.03** |
| Nguyen & Ho (2024) (LCV) | 0.735 | 0.218 | 1.81 |
| Nguyen & Ho (2024) (UCV) | 0.726 | 0.200 | 2.19 |
| Deshpande et al. (2019) (Max-SW) | 29.152 | **0.0497** | 1.13 |
| Nguyen & Ho (2023) (IS-EBSW-e) | 0.698 | 0.2291 | 1.42 |
| Nguyen et al. (2024a) (QMC) | 0.670 | 0.208 | 1.38 |
| **ReSWD** | **0.622** | 0.278 | 1.92 |
| **ReSWD + QMC** | **0.610** | **0.274** | 2.10 |

Table 2: **Influence of fresh candidates.** With a fixed budget of 64 projections, we analyze the influence of the fresh candidates.

| # Candidates | Mean-$W_1$ $[10^3]$ ↓ | Time per step [ms] ↓ |
|---|---|---|
| 2 | 0.721 | 1.99 |
| 4 | 0.673 | 1.96 |
| **8** | **0.622** | 1.92 |
| 16 | 0.746 | **1.91** |
| 32 | 1.192 | 1.98 |
| 48 | 2.122 | 1.93 |
| 56 | 3.811 | **1.85** |

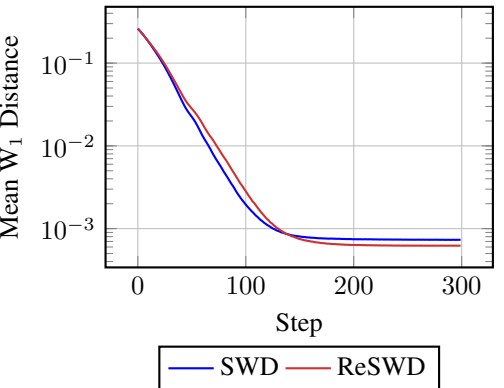

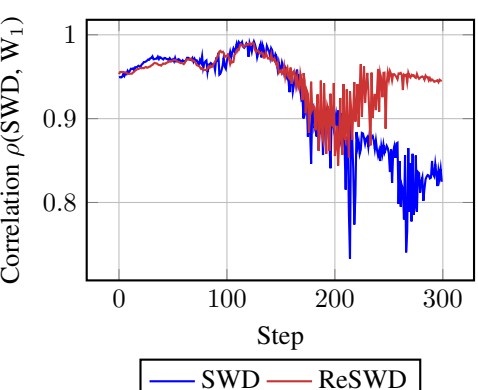

Figure 3: **True Wasserstein metric over steps.** The effect of reservoir warmup is clear when comparing ReSWD with the true Wasserstein distance during optimization. Initially, ReSWD performs slightly worse due to lower projections in the loss, but can outperform SWD in the end.

Figure 4: **Pearson correlation with the true Wasserstein Distance.** Our method achieves a high correlation with the true Wasserstein loss, while improving upon pure SWD (See Fig. 3). This indicates the unbiased nature of our proposed method.

Table 3: **Comparison on color matching.** Here, we compare ReSWD with a baseline SWD method and prior color matching works by Reinhard et al. (2001), Nguyen et al. (2014) as well as neural network based methods by Yoo et al. (2019), Ho & Zhou (2021) and Larchenko et al. (2025). We report the errors between the color charts of adjusted and ground truth images as PSNR, transform and CDL error and the image quality as CTQM. Additionally, we compare the runtimes of the methods.

| Method | Color PSNR ↑ | Transform err. (RMSE) ↓ | CDL err. (RMSE) ↓ | CTQM ↑ | Time per match [s] ↓ |
|---|---|---|---|---|---|
| Reinhard et al. (2001) | 21.94 | **0.31** | 0.14 | 5.12 | **1** |
| Nguyen et al. (2014) | 18.76 | 1.27 | 0.33 | 5.16 | 3 |
| Yoo et al. (2019) | 20.97 | 0.44 | 0.21 | 5.09 | 20 |
| Ho & Zhou (2021) | 10.43 | 0.54 | 0.45 | 5.11 | 2 |
| Larchenko et al. (2025) | 14.80 | 0.47 | 0.20 | 5.05 | 24 |
| Ours with SWD | 24.30 | 0.34 | 0.11 | 5.15 | 5 |
| **ReSWD** | **24.64** | **0.31** | **0.10** | **5.17** | 5 |

**Diffusion Guidance.** We select Lobashev et al. (2025) as our main comparison. Furthermore, we also implemented our modifications to the matching in the SDXL model to compare our alterations with the baseline, which is based on SDXL. This delineates the influence of the improved SD3.5 model from our matching and alterations. For evaluations, we follow Lobashev *et al.* and select 1000 prompts from the ContraStyles dataset[1] and 1000 images from Unsplash Lite[2]. We provide the $W_2$ Wasserstein distance, CLIP-IQA (Wang et al., 2023) and CLIP-T (Radford et al., 2021) to present color matching performance, highlight the quality of the generation and prompt adherence, respectively.

---

[1] https://huggingface.co/datasets/tomg-group-umd/ContraStyles
[2] https://unsplash.com/data

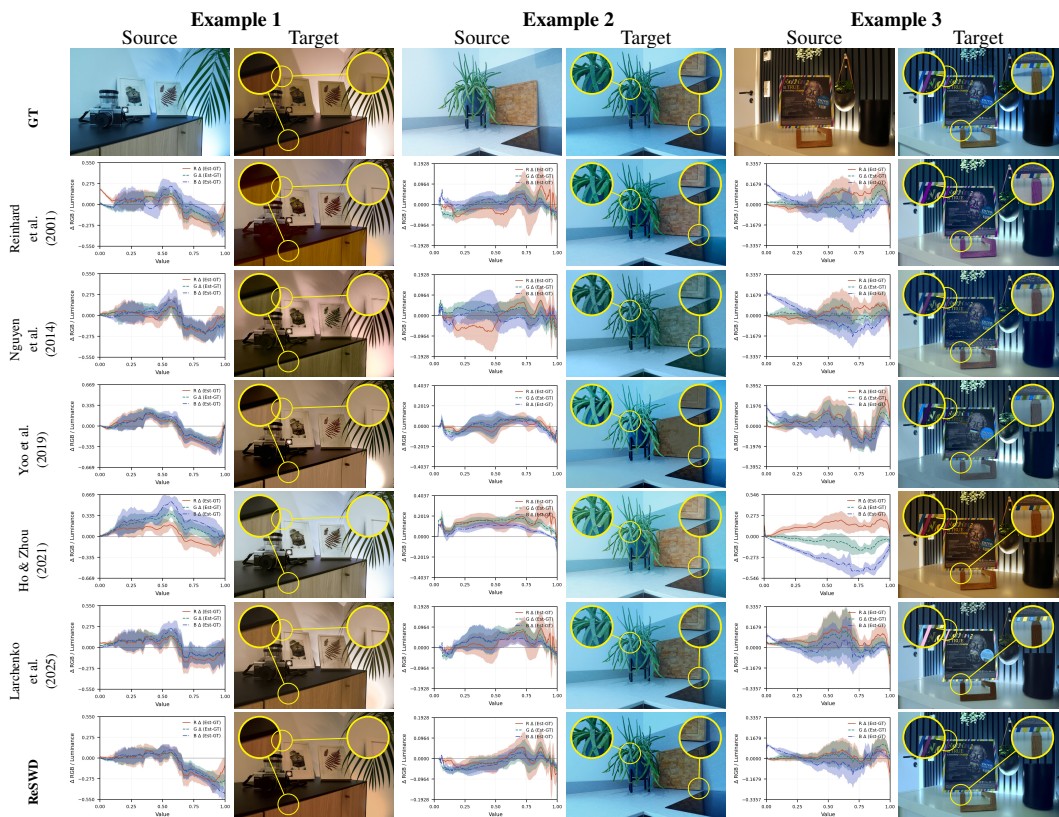

Figure 5: **Color Matching.** Notice the consistent images our method produces with accurate shot matching. ReSWD consistently produces good results which match the final color distribution and without introducing any artifacts. We highlighted challenging areas which our method handled well. This is also evident by comparing the delta with the ground truth for each color channel. Ours produces fewer deviations with less variance. Note that the reference implementations of Larchenko et al. (2025), Yoo et al. (2019) and Nguyen et al. (2014) do not support high resolution images.

Table 4: **Comparison on diffusion guidance.** Here, we compare ReSWD with Lobashev et al. (2025) on color guidance in image generation. As our main proposed method uses the more recent SD3.5, we also compare with our modifications on SDXL. Even on the same base model our modifications improve upon the base technique.

| Method | CLIP-IQA ↑ | CLIP-T ↑ | Mean-$W_2$ [$10^2$] ↓ | Time per generation [s] ↓ |
|---|---|---|---|---|
| Lobashev et al. (2025) | 0.671 | 14.861 | 1.941 | 124 |
| ReSWD w/o Adam & GradStop @ SDXL | 0.691 | 14.848 | 1.832 | 126 |
| ReSWD w/o Adam @ SDXL | 0.681 | 14.771 | 1.846 | 34 |
| ReSWD @ SDXL | 0.696 | 14.870 | 1.213 | 34 |
| ReSWD @ SD3.5-medium | 0.786 | 14.783 | 0.815 | 32 |
| ReSWD @ SD3.5-turbo | **0.800** | **14.882** | 0.675 | **4** |
| ReSWD @ SD3.5-large | 0.793 | 14.817 | **0.55** | 69 |

In Table 4 our modifications clearly outperform the prior state-of-the-art technique of Lobashev *et al*. Even with the same base model (SDXL), our method is faster and provides better results, and this holds true with additionally ablating the optimizer improvements and without the gradient stopping. As seen here, our gradient stopping implementation does not influence on the guidance negatively but creates a large speed-up as well as a significant memory reduction which allows us to use larger base models. With the upgraded base model, ReSWD clearly outperforms the previous method. Here, it is interesting that the distilled Turbo variants provides better quality, which we attribute to the straighter trajectories.

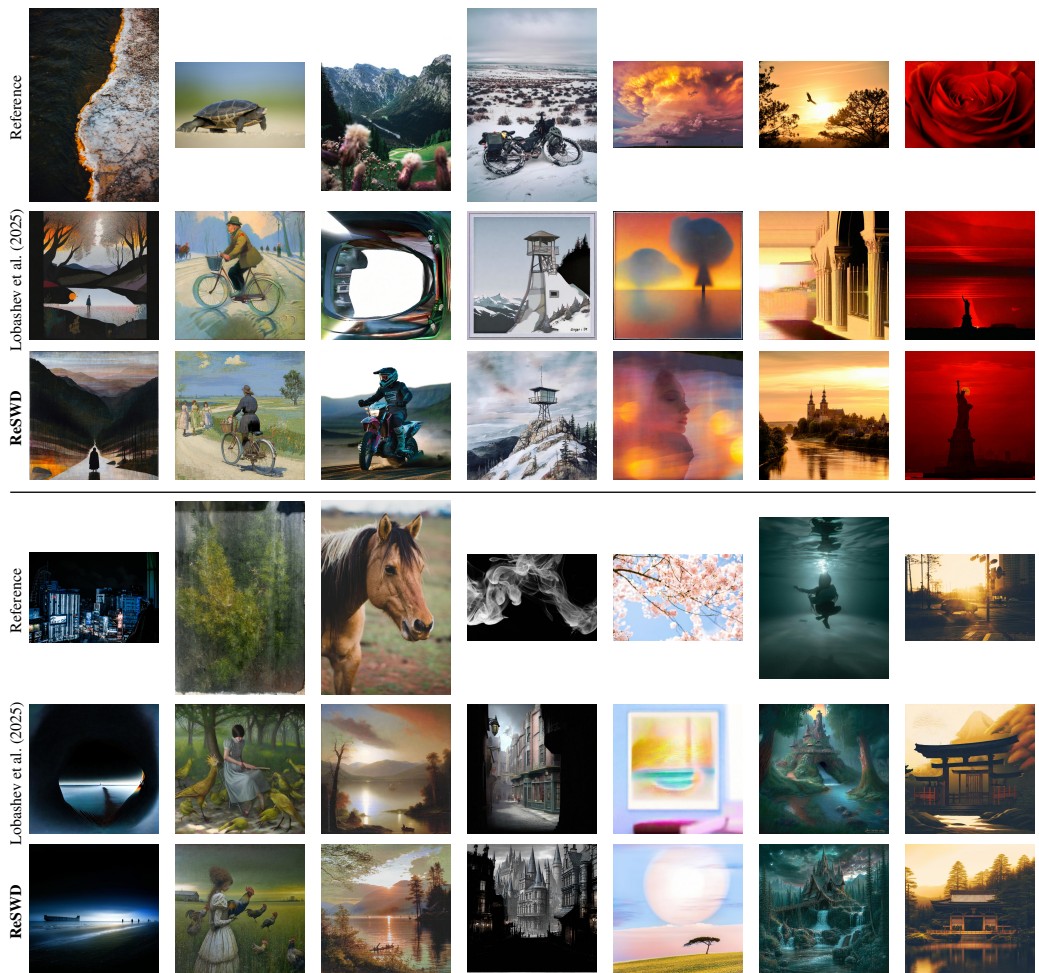

Figure 6: **Diffusion Guidance.** Visualization of our color guidance with SD3.5-Large. Notice, how closely the colors match the reference and the detailed generation quality of our method. Also we noticed that Lobashev et al. (2025) sometimes degenerates (Third example top, First example bottom) whereas ReSWD handles these challenging guidance signal well.

Table 5: **Influence of dimensions.** Increasing the dimensionality of the problem introduces a decrease in quality. This comes from the fact that the reservoir mechanism becomes less likely to receive good candidates as randomly sampling them becomes unlikely; thus, the reservoir is not updated as often, and the direction becomes stale.

| Method | Mean-$W_1$ [$10^3$] $\downarrow$ | | | | | | |
| --- | --- | --- | --- | --- | --- | --- | --- |
| | 3 | 6 | 12 | 24 | 32 | 48 | 64 |
| SWD | 0.733 | 0.945 | 1.190 | 3.386 | 4.863 | 6.503 | 7.922 |
| Nguyen & Ho (2024) (LCV) | 0.735 | 0.963 | 2.203 | 4.186 | 5.271 | 7.152 | 8.826 |
| Nguyen & Ho (2024) (UCV) | 0.726 | 0.941 | 1.935 | 3.815 | 4.919 | 6.573 | 8.009 |
| Deshpande et al. (2019) (Max-SW) | 29.152 | 72.572 | 127.321 | 172.492 | 186.748 | 201.986 | 210.266 |
| Nguyen & Ho (2023) (IS-EBSW-e) | 0.698 | 0.887 | 1.536 | **2.812** | **3.518** | **4.774** | **6.044** |
| Nguyen et al. (2024a) (QMC) | 0.670 | **0.736** | **1.166** | 3.904 | 4.899 | 6.509 | 7.854 |
| **ReSWD** | 0.622 | 0.939 | 9.768 | 15.350 | 17.114 | 19.578 | 21.446 |
| **ReSWD + QMC** | **0.615** | 0.823 | 4.532 | 8.953 | 14.389 | 18.375 | 20.862 |

We show example generations of this dataset in Fig. 6 along with the reference and the prompts. As evident, the improved generation capacity of SD3.5 along with our necessary changes resulted in drastically better generation which are faithful to the color distribution defined by the input image.

**Influence of Dimensionality.** Our method is designed to maintain good directions in a reservoir that will be reused during optimization in an unbiased way. However, with higher dimensionalities the chance of drawing a good kernel is drastically lowered. Therefore, the reservoir will not be updated and the directions become stale. This effect can be seen in Table 5, where our solution drastically degrades in a 12 dimension problem. This could be circumvented with more aggressive ESS clearings or time-based decay, but at this point our method would revert back to plain SWD. We can also combine our method with QMC Nguyen et al. (2024a) to alleviate some of the issues, but even with that the chance of sampling good filters is drastically decreased.

**Ablation.** In Table 2, we assess the influence of the number of new candidates in our general distribution matching task. Too few new candidates starve the reservoir of new directions, resulting in static directions during optimization. Too many new candidates reduce the reservoir size, which drives the optimization solely, resulting in too few directions supporting the optimization. With 8 candidates, we achieved the best overall results in all applications given 64 total projections.

**Limitations.** Our current technique is limited to simple matrix projections, and an extension to learned convolution kernels similar to Elnekave & Weiss (2022) did not provide satisfactory results, as similar to the dimensionality problem for 1D distributions, the search space for kernels is too large to randomly obtain drastically better kernels. This degraded the results similar to limiting the projection directions and reducing the redrawing of new directions to every few optimization steps.

## 5 CONCLUSION

Our novel combination of real-time rendering-influenced variance reduction techniques in SWD optimization offers a more efficient and unbiased solution compared to other recent variance reduction techniques. This results in reduced overhead while maintaining optimal performance. Our technique has demonstrated superior performance in various real-world and synthetic applications, achieving state-of-the-art results.

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

## SUPPLEMENTS

In the supplements, we present more results from our applications.

## A  DIFFUSION GUIDANCE

In Fig. 7 we present further results on diffusion guidance. Lobashev et al. (2025) proposed method and implementation often collapse, blur, or contain artifacts when the SWD guide enforces unlikely colors given the prompt. Our updated pipeline does not produce these issues.

## B  COLOR MATCHING

We show additional results in Fig. 8 with our baselines. Notice that our results achieve consistent color matching performance and remain artifact-free. Our method also supports running at an arbitrary resolution.

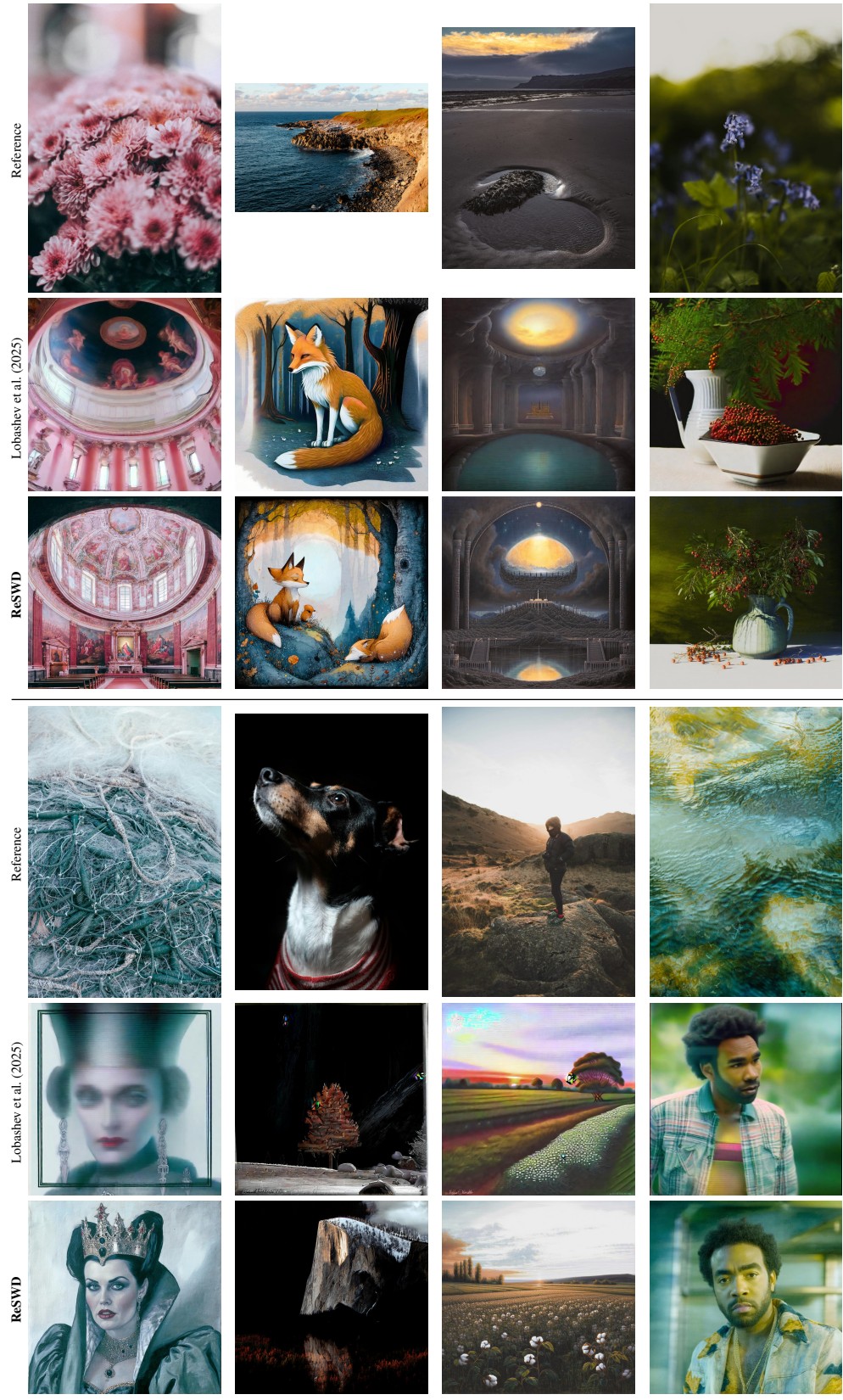

Figure 7: **Diffusion Guidance.** Further results on diffusion guidance compared to Lobashev et al. (2025). Notice their approach often becomes blurry, collapses or contains artifacts.

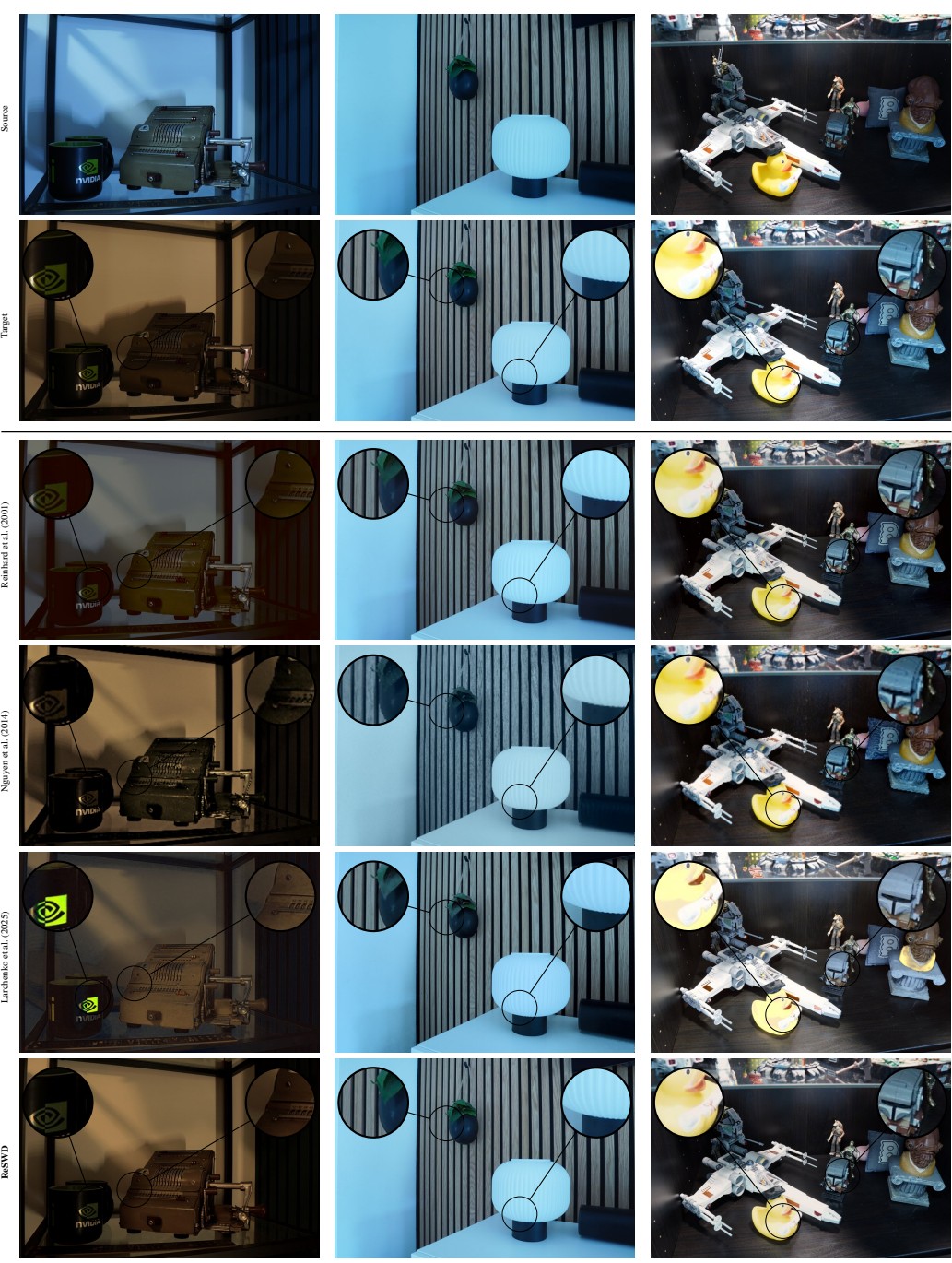

Figure 8: **Color Matching.** More results on the color matching task. Notice the subtle changes and artifact free, high resolution color matching our method enables.

