# OpenReview forum: "ReSWD: ReSTIR‘d, not shaken. Combining Reservoir Sampling and Sliced Wasserstein Distance for Variance Reduction."
_ICLR.cc/2026/Conference — ICLR 2026 Conference Desk Rejected Submission_

### Official Review · Reviewer_a2rM · 2025-10-19

**Soundness:** 3
**Presentation:** 3
**Contribution:** 3
**Rating:** 8
**Confidence:** 5

**Summary:**

The paper “ReSWD: ReSTIR’d, Not Shaken” introduces Reservoir Sliced Wasserstein Distance (ReSWD), a variance-reduced and unbiased variant of the Sliced Wasserstein Distance (SWD) for efficient distribution matching in high-dimensional settings. By integrating Weighted Reservoir Sampling (WRS), a technique inspired by real-time rendering (ReSTIR), ReSWD adaptively retains projection directions that contribute most to the loss, leading to smoother gradients and faster convergence without bias. The method employs time-decay weighting, effective sample size control, and reuses informative directions across optimization steps. Experiments on synthetic benchmarks, color correction, and diffusion model guidance demonstrate that ReSWD consistently outperforms standard SWD and existing variance reduction approaches, improving both accuracy and computational stability.

**Strengths:**

1. Novel integration of rendering and optimal transport: ReSWD creatively bridges computer graphics (ReSTIR) and machine learning (SWD), introducing a new, cross-disciplinary variance reduction method.

2. ReSTIR provides stable and low-variance gradients while remaining unbiased, directly improving optimization speed and robustness.

3.  Across both synthetic and real-world tasks (color correction and diffusion guidance), ReSWD shows consistent and measurable performance gains over SWD and prior baselines.

4. The method adds minimal computational overhead compared to SWD but achieves substantial improvements in convergence and stability.

5. ReSWD can be plugged into any optimization pipeline involving SWD, making it useful for computer vision, graphics, and generative modeling tasks.

6.  The reservoir sampling framework provides an intuitive way to focus computation on the most informative directions, improving interpretability and control.

**Weaknesses:**

1. Dependency on hyperparameters: The performance relies on careful tuning of parameters such as reservoir size, number of new directions, and time-decay constant, which may vary across tasks.

2. Initial warm-up lag: During early optimization, ReSWD may perform slightly worse than standard SWD until the reservoir becomes well populated, introducing a short initial slowdown.

3. Added implementation complexity: While computational cost is low, integrating the reservoir update and WRS mechanism adds code complexity compared to the simplicity of vanilla SWD.

**Questions:**

1. Can we use Quasi-Monte Carlo for reservoir construction? It seems that Weighted Reservoir Sampling  is an orthogonal techniques to existing variance reduction approaches for SWD. If the author agrees with my comments, some discussion should be added to the paper.

2. Sliced Wasserstein distance can be computed for unequal sample sizes (see Section 2.3 in [1]). Why, then, is it necessary to make the sample sizes equal? This approach may lead to faster computation; however, I wonder whether the performance of applications would change if we did not resample to obtain equal sample sizes. The authors should also mention that the Sliced Wasserstein distance can be computed for any discrete distributions after equation (1) and cite the relevant works to avoid potential misunderstandings about the applicability of SWD.

3. Self-normalized weights are similar to the Sampling Importance Resampling approach in energy-based sliced Wasserstein [2]. The authors wrote that "This remains an unbiased MC estimate of $\mathbb{E}_\theta [W_p]$" (line 228). Is $\theta$ is still uniform distributed?

[1] "An Introduction to Sliced Optimal Transport", Khai Nguyen

[2] "Energy-based Sliced Wasserstein Distance", Nguyen et al.

---

> ### Author Response · Authors · 2025-11-18
>
> We thank the reviewer for the suggestions. We introduced the following changes in the paper:
>
> ## Weakness 1:
>
> For the other parameters, we did not find any specific sensitivities. The number of directions has a similar effect to all SWD papers. Increasing the directions improves the results but becomes expensive. Too few directions, and the matching performance degrades. Similar to previous papers, we found 64 a good compromise. A too-low ESS alpha triggers the reset often, which degrades the method to SWD with M fewer filters. We found 0.5 a threshold which rarely triggers and prevents a collapse in rare cases.
>
> ## Weakness 2:
>
> This is true, and we showcase the behavior in Figure 3 and 4. However, given that we do not require any inner loop optimization for the loss calculation, we believe this is a worthwhile trade-off. As seen, we catch up with SWD quickly and outperform it.
>
> ## Weakness 3:
>
> The integration and connection of these techniques are not obvious, and even the integration requires some novelty. In a similar sense, the integration of QMC into SWD or Control Variates would then also warrant the same critique, but they are some of the best-performing techniques in the field.
>
> ## Question 1:
>
> Yes, we can combine ReSWD and QMC for further improvements. We have added it to Table 1 in the revision.
>
> | Method                          | Mean-W₁ [10³] ↓ | Grad SNR ↑ | Time per step [ms] ↓ |
> |---------------------------------|------------------|-------------|------------------------|
> | SWD                             | 0.733            | 0.215       | 1.03                   |
> | Nguyen et al. (2024) (LCV)      | 0.735            | 0.218       | 1.81                   |
> | Nguyen et al. (2024) (UCV)      | 0.726            | 0.200       | 2.19                   |
> | Deshpande et al. (2019) (Max-SW)| 9.740            | 0.286       | 7.21                   |
> | Nguyen et al. (2024) (QMC)      | 0.670            | 0.209       | 1.51                   |
> | **Ours**                        | 0.622            | 0.278       | 1.92                   |
> | **Ours + QMC**                  | 0.610            | 0.274       | 2.10                   |
>
> ## Question 2:
>
> We have clarified the handling of varying sample counts in the paper. SWD requires an equal number of samples. The duplication is a common mechanism to relax this requirement.
>
> ## Question 3:
>
> Our new samples are still drawn from a uniform distribution. Otherwise, the PDF of drawing the directions has to be accounted for, which is possible in the WRS framework.

---

> > ### Comment · Reviewer_a2rM · 2025-11-19
> >
> > Thank you very much for your response. Could you please elaborate more on Question 3? I'm still not clear about "This remains an unbiased MC estimate of $E_\theta [W_p]$, while focusing on the computation on projections that highlight larger differences. "

---

> > > ### Author Response · Authors · 2025-11-20
> > >
> > > Instead of randomly sampling points everywhere with equal likelihood (like in SWD), our approach is inspired by importance sampling. We sample more often in places where the function is "important" (large contribution). This "overweights" the important regions. We correct for the fact that with the weights from the WRS. This weight compensates for how much more (or less) frequently you sampled there compared to what you would if sampling randomly. Hence, remaining unbiased but reducing variance. We will clarify this further.

---

### Official Review · Reviewer_3dAY · 2025-10-28

**Soundness:** 2
**Presentation:** 3
**Contribution:** 2
**Rating:** 2
**Confidence:** 4

**Summary:**

Sliced Wasserstein estimates enable comparison of high-dimensional distributions and are often used as loss functions in gradient-based optimization. Their computation relies on Monte Carlo approximation through random projections. In recent years, there has been growing interest in the 'smart' selection of projection directions. The authors contribute to this line of work by introducing a Monte Carlo sampling scheme tailored for optimization that reuses projection directions found to be informative during training.

**Strengths:**

- The key idea of keeping `good’ directions for SWD gradient descent seems intuitive and useful.
- The key ideas are well-explained and the visualized.
- The paper is well-written and the key challenge is clear.
- The experiments are suitable for the method in question
- The authors added a nice explanatory video in the supplementary material

**Weaknesses:**

- The reported quantitative improvements over standard SWD are very minor.

- While the authors mention alternatives to standard Monte Carlo SWD, they mostly do not compare against it. There exist numerous spherical slicing schemes that should be compared against, see
	- Quasi-monte carlo for 3d sliced Wasserstein (Nguyen et al, 2023)
	- Fast summation of radial kernels via QMC slicing (Hertrich et al, 2024)
	- Energy-based sliced Wasserstein distance (Nguyen & Ho, 2023)
	- A User's Guide to Sampling Strategies for Sliced Optimal Transport (Sisouk et al, 2025)
	- MaxSliced Wasserstein, Repulsive Monte Carlo, ….

- Generally, I would not consider color matching as the `ideal’ application for such experiments as the success mostly depends on the color palette, e.g., “RGB” vs “HGL” vs “HSV” et.

- The paper relies heavily on a few visual examples, which take up a lot of space. However, visual differences between most methods are very minor.

- The main idea is mainly a straightforward application of the ideas in (Efraimidis & Spirakis, 2006)

- Code is missing

Conclusion:
While the idea is intuitive and potentially useful, I feel as if the reported experiments do not support the clear advantage of the method. Moreover, the extensive research on the 'smart' selection of slicing directions has not really been covered.

**Questions:**

- Figure 2 shows “Calculate Loss $W_p^p$. Do you mean `Calculate Loss SWD’? Both formulations make sense, but it might be useful to add “1D $W_p^p$” to avoid confusion if it actually is $W_p^p$”.

- In Section 3, the cardinality of X is set to $n_X$ and $n_Y$ for Y, but in Eq. (1) it is assumed that $n=n_X =n_Y$. If $n_X\neq n_Y$ a more general formula is needed.

- The statement ‘The SWD is an unbiased estimate of the true Wasserstein distance’ (line 159) is incorrect. I assume that you refer to the true SWD distance and the numerical estimator?

---

> ### Author Response · Authors · 2025-11-18
>
> We thank the reviewer for the suggestion. We enumerated the bullet points for easier reference. We introduced the following changes in the paper:
>
> ## Weakness 1/2:
>
> Our method outperforms most major additions to SWD by a significant margin. For example, Control Variates, QMC, and Max-SW. The improvements from our method have a significantly higher delta to SOTA methods. Furthermore, we can combine our method with QMC as it is orthogonal to our method. This improves the performance even more.
>
> | Method                          | Mean-W₁ [10³] ↓ | Grad SNR ↑ | Time per step [ms] ↓ |
> |---------------------------------|------------------|-------------|------------------------|
> | SWD                             | 0.733            | 0.215       | 1.03                   |
> | Nguyen et al. (2024) (LCV)      | 0.735            | 0.218       | 1.81                   |
> | Nguyen et al. (2024) (UCV)      | 0.726            | 0.200       | 2.19                   |
> | Deshpande et al. (2019) (Max-SW)| 9.740            | 0.286       | 7.21                   |
> | Nguyen et al. (2024) (QMC)      | 0.670            | 0.209       | 1.51                   |
> | **Ours**                        | 0.622            | 0.278       | 1.92                   |
> | **Ours + QMC**                  | 0.610            | 0.274       | 2.10                   |
>
>
> ## Weakness 3:
>
> We also showcased a generic distribution matching task, where our method also outperforms existing ones. Here, no color space is influencing the results. Our applications contain additional key contributions that improve over any SOTA method in their field.
>
> ## Weakness 4:
>
> While the differences in color matching are minor, they can be noticed and it is evident that our method does not introduce any artifacts or severe color shifts as seen in other methods. For example, in Fig. 5 the record and wood stand. Existing solutions either flip the color, boost the intensity or produce too dull results. To highlight this, we added color delta plots per scene to Fig. 5. Another example is in the second example, where a white halo is produced next to the picture frame, and the wood of the cabinet becomes too orange.
>
> ## Weakness 5:
>
> The integration and connection of WRS and SWD is not obvious and it is novel. It is not a straightforward application of WRS on SWD. In a similar sense the integration of QMC or Control Variates into SWD would also warrant the same critique, but they are some of the highest performing techniques in the field and introduced variance reduction to SWD.
>
> ## Weakness 6:
>
> The code of the paper including demo applications will be released.
>
> ## Questions:
>
> - We have clarified the loss in Figure 2.
> - We describe handling of unequal sample counts in paragraph “Handling unequal sample counts”. We have now added a reference to it in the algorithm.
> - We calculate the true Wasserstein Distance. If we would compare it with the sliced distance, then SWD would have a correlation of 1.

---

> > ### Comment · Reviewer_3dAY · 2025-11-21
> > **Reply**
> >
> > Hello,
> >
> > I thank the authors for their answers! Again, the paper is well-written and the methods seems to be perform well for the tasks at hand. However, my concerns persist.
> >
> > 1. If the main application is color transfer, there should be more comparisons with non-OT-based methods as pointed out by other reviews. As an example, if my main focus is quick color transfer in video streaming applications, I would assume that it is faster to train a shallow UNet/CNN.
> >
> > 2. Moreover, the proposed method does not seem to scale to high-dimensional settings. However, high-dimensional cases are a crucial weakness of slicing.
> >
> > 3. While the authors added comparative results to some alternative methods, there are a lot more, see my original review. Given the amount of methods, I do not feel as if the literature review and the benchmarking are up to ICLR-standards.
> >
> > 4. In low-dimensional settings (eg color), max-slicing can be done using the maximum Monte Carlo protection, see the POT implementation. Therefore, I am surprised at the presented max-slicing runtime.
> >
> > 5. QMC directions can be computed offline and just randomly rotated during optimization, see some of the references. In this case, QMC should not take longer than Monte Carlo SW. Thus, I am again surprised at the runtime.
> >
> > 6. It would have been easier for reviewers to investigate 4./5. if the code had been provided to us.
> >
> > 7. If one Monte Carlo SW optimization step takes half as long as current implementation, I might as well do twice as many steps, couldn't I?
> >
> > I think that earlier QMC work on slicing has been valuable and that the proposed approach might be valuable. However, I feel as if the current paper does not really build on earlier research. As an example, it would have been interesting to apply your methods to other sliced divergences, e.g., sliced MMD. Overall, I think that this is an interesting approach, but that the experimental and methodological focus should have been broader.

---

> > > ### Author Response · Authors · 2025-11-27
> > >
> > > Thank you for the insightful comment. Regarding the issues addressed in the answer:
> > >
> > > 1.	We compare with non-OT methods on the color matching task in Table 3. Larchenko is a SOTA neural network-based technique. Reinhard and Nguyen are more traditional non-SOTA-based matching methods. We have added additional comparisons of neural network-based methods to the table (Ho et al. & Yoo et al.). In the visual comparisons in Fig. 5, ours also outperforms the existing methods.
> > >
> > > | Method                                      | Color PSNR ↑ | Transform err. (RMSE) ↓ | CDL err. (RMSE) ↓ | CTQM ↑ | Time per match [s] ↓ |
> > > |---------------------------------------------|--------------|---------------------------|---------------------|--------|------------------------|
> > > | Reinhard et al. (2001)                      | 21.94        | 0.31                      | 0.14                | 5.12   | 1                      |
> > > | Nguyen et al. (2014)                        | 18.76        | 1.27                      | 0.33                | 5.16   | 3                      |
> > > | Yoo et al. (2019)                           | 20.97        | 0.44                      | 0.21                | 5.09   | 20                     |
> > > | Ho et al. (2021) (DeepPreset)               | 10.43        | 0.54                      | 0.45                | 5.11   | 2                      |
> > > | Larchenko et al. (2025)                     | 14.80        | 0.47                      | 0.20                | 5.05   | 24                     |
> > > | Ours with SWD                                | 24.30        | 0.34                      | 0.11                | 5.15   | 5                      |
> > > | **Ours**                                     | 24.64        | 0.31                      | 0.10                | 5.17   | 5                      |
> > >
> > >
> > > 2. Yes, our method does indeed seem limited in high-dimensional cases. We do not claim that we tackle or improve this problem. However, we increased the performance on many lower-dimensional problems significantly and proposed two applications with several novel additions. These implementations outperform all SOTA methods.
> > >
> > > 3. Several of the works cited here only discuss QMC methods, which we evaluated against. We’ve additionally added “Energy-Based Sliced Wasserstein Distance” from Nguyen et al. Ours outperforms their method as well on the general matching.
> > >
> > >
> > >
> > > | Method                                   | Mean-W₁ [10³] ↓ | Grad SNR ↑ | Time per step [ms] ↓ |
> > > |------------------------------------------|------------------|-------------|------------------------|
> > > | SWD                                      | 0.733            | 0.215       | 1.03                   |
> > > | Nguyen et al. (2024) (LCV)               | 0.735            | 0.218       | 1.81                   |
> > > | Nguyen et al. (2024) (UCV)               | 0.726            | 0.200       | 2.19                   |
> > > | Deshpande et al. (2019) (Max-SW)         | 29.152           | 0.0497      | 1.13                   |
> > > | Nguyen et al. (2023) (IS-EBSW-e)         | 0.698            | 0.2291      | 1.42                   |
> > > | Nguyen et al. (2024) (QMC)               | 0.670            | 0.208       | 1.38                   |
> > > | **Ours**                                 | 0.622            | 0.278       | 1.92                   |
> > > | **Ours + QMC**                           | 0.610            | 0.274       | 2.10                   |
> > >
> > >
> > >
> > > 4. We implemented Max Slicing as an inner loop optimization of the directions. This implementation was also used in the “Energy-Based Sliced Wasserstein Distance” paper. We have attached the runtime cost and performance of using the POT implementation. The results degraded significantly. See the results in the above table.
> > >
> > > 5. We have updated the implementation using orthonormal rotation matrices as per the official implementation. The method achieves the same results but now with a lower runtime of 1.38ms per optimization step. We have added the results in the table above and in the paper.
> > >
> > > 6. The code will be released soon. Per conference and reviewer guidelines, code upload is not required. Due to our lab's situation, and many others, code release is a slower process that we are still working on. We have provided pseudo-code that clearly defines how the algorithm is implemented.
> > >
> > > 7. For tasks where the evaluation is cheap, this holds true (Color Matching, General Distribution Matching). However, in our diffusion color guidance, the evaluation of the networks dominates the optimization cost. Here, it would not be feasible to double the optimization steps. We have additionally done an experiment where we increased the optimization of SWD to 600 from 300, and the final metric only improved slightly to 0.730. Furthermore, in Fig. 3, it can be seen that the loss curve becomes flat around step 200.

---

### Official Review · Reviewer_d1cU · 2025-10-31

**Soundness:** 3
**Presentation:** 3
**Contribution:** 2
**Rating:** 4
**Confidence:** 3

**Summary:**

The n authors noted that while various variance reduction techniques exist for Monte Carlo (MC) estimators, they are not widely used for distribution matching objectives. They created a new estimator called Reservoir SWD (ReSWD). They  draw inspiration from a rendering technique called ReSTIR and a sampling algorithm called Weighted Reservoir Sampling (WRS) and adapt WRS to the Sliced Wasserstein Distance (SWD) setting.

**Strengths:**

1) Their method reduces stochastic variance, preserves unbiasednes and leads to faster and more robust optimization.

2)  They demonstrated the advantages of ReSWD on both real-world applications (color correction and color diffusion guidance) and synthetic distribution matching problems, showing it outperforms the standard SWD and other baseline methods

3) Their method demonstrates the scalability

**Weaknesses:**

1) The method concentrates on directions where the distributions are "most dissimilar." There is a risk that the optimizer could over-optimize for these specific directions in the reservoir at the expense of the overall distribution match.

2)The method concentrates on directions where the distributions are "most dissimilar." There is a risk that the optimizer could over-optimize for these specific directions in the reservoir at the expense of the overall distribution match.

3)The text emphasizes that the estimator preserves unbiasedness. This is a theoretical strength for the value of the distance estimate. However, in optimization, what matters most are the gradients?

4) The method seems more complex than standard SWD. It requires correctly implementing the WRS algorithm and integrating it with the gradient update process.

**Questions:**

1) The WRS process adds per-step overhead. Was there a measurable trade-off where this overhead outweighed the benefits of variance reduction, especially in the early stages of optimization or for very simple problems?


2) Why choose WRS over other variance reduction techniques like control variates or stratified sampling adapted for this setting?

---

> ### Author Response · Authors · 2025-11-18
>
> We thank the reviewer for the suggestion. We introduced the following changes in the paper:
>
> ## Weakness 1/2:
>
> Our optimizer does not influence the choice of the directions in the reservoir. These are selected by an algorithm based on the current loss landscape. As this happens at every distribution step and we implement time-decay and ESS mechanisms, a collapse as discussed is avoided.
>
> ## Weakness 3:
>
> We have added the influence on gradients to Table 1. As seen there, our methods provide a drastically improved SNR compared to SWD, but it is weaker than the one from Max-SW. However, the optimization result from Max-SW is not as good as ours, and the optimizing takes significantly longer. Intuitively, this comes from replacing the mean with a max estimator and only backpropagating through a single kernel. The added variance from visiting more features seems to be helpful in these settings. Our method strikes a balance between SWD and Max-SW by learning a reservoir of multiple meaningful directions with minimal overhead.
>
> | Method                          | Mean-W₁ [10³] ↓ | Grad SNR ↑ | Time per step [ms] ↓ |
> |---------------------------------|------------------|-------------|------------------------|
> | SWD                             | 0.733            | 0.215       | 1.03                   |
> | Nguyen et al. (2024) (LCV)      | 0.735            | 0.218       | 1.81                   |
> | Nguyen et al. (2024) (UCV)      | 0.726            | 0.200       | 2.19                   |
> | Deshpande et al. (2019) (Max-SW)| 9.740            | 0.286       | 7.21                   |
> | Nguyen et al. (2024) (QMC)      | 0.670            | 0.209       | 1.51                   |
> | **Ours**                        | 0.622            | 0.278       | 1.92                   |
> | **Ours + QMC**                  | 0.610            | 0.274       | 2.10                   |
>
>
> ## Weakness 4:
>
> Every modification of a base algorithm adds some complexity. Ours adds little overhead in runtime and is mostly straightforward to implement without any inner optimization loops.
>
> ## Question 1:
>
> The overhead on the loss is not too large. For example, in the diffusion guidance approach, the introduction of ReSWD without any other introductions increases the run-time from 124s to 126s, which is roughly a 1% runtime increase, while still delivering better results. A single extra optimization step would take longer and would not influence the results enough to overcome this.
>
> ## Question 2:
>
> Control Variates (CV) have already been explored by Nguyen & Ho (2024), and in our applications, our method outperforms CV. Quasi Monte Carlo approaches are similarly explored by Nguyen et al. (2024a) and can be combined with our approach as they are orthogonal when properly taking the reservoir weights into account.

---

### Official Review · Reviewer_Kxjt · 2025-11-01

**Soundness:** 3
**Presentation:** 4
**Contribution:** 3
**Rating:** 4
**Confidence:** 4

**Summary:**

This paper combines Weighted Reservoir Sampling (WRS) and the Sliced Wasserstein Distance (SWD) to propose ReSWD. SWD, introduced to mitigate the curse of dimensionality, still suffers from noisy gradients and slow convergence when used as a loss function in optimization scenerios. This drawback is caused by the high variance of SWD's Monte Carlo estimator. By adaptively retaining the projection directions that are most dissimilar between the two distributions, ReSWD significantly reduces variance while preserving unbiasedness. The experiments presented in the paper show that ReSWD not only surpasses the standard SWD across various tasks, but also outperforms previous SOTA benchmarks on real-world tasks such as color matching and color-guided diffusion image generation.

**Strengths:**

1. ReSWD addresses real-world color matching problems from the perspective of distribution matching, providing a more data efficient and stable solution to such applications. ReSWD enables the authors to tackle the color matching problem using the Color Decision List, which greatly improves controllability for subsequent manual edits.

2. The paper incorporates good practices from prior works, such as Bitterli et al., 2020 and Elnekave & Weiss (2022), facilitating ReSWD's ESS-based reservoir resets and handling of unequal sample counts.

3. The paper is clearly and thoroughly presented. Section 3.1 includes a concise yet precise definition of the problem and highlights the limitations of SWD. Section 3.2 introduces the proposed method in a well-structured, step-by-step manner, accompanied by a clear and detailed pseudocode for its algorithm.

**Weaknesses:**

1. Table 1 and 3 show the superiority of ReSWD over SWD. However, the performance gap between the two methods seems relatively small, especially for real-world tasks like color matching. It would be very helpful for readers to better assess the effectiveness of ReSWD if visual results of the standard SWD are given in Figure 5 and 6.

2. Both line 217 of the main text and line 11 of Algorithm 1 state that the *smallest* K keys should be kept. However, based on the definition for $k(\theta)$ given in the paper and the fact $u \sim \mathcal{U}(0,1)$, it appears to me that the *largest* K keys should be kept. I would be glad to revise my assessment if the authors can clarify this point and demonstrate why the smallest keys are correct in this context.

3. According to Algorithm 1, it appears time-decay reweighted keys $\tilde{k_i}$ are not actually used and are overwritten at line 9. If my interpretation is incorrect, I would appreciate clarification from the authors. Again, I would be glad to revise my assessment in that case.

4. The last row (# Candidates = 56) shows the lowest time per step (1.85s) in Table 2. However 1.91ms and 1.92 ms are marked as fastest instead.

5. If I understand correctly, Figures 6 shows ReSWD outputs generated with SD3.5 while the results of Lobashev et al. (2025) are generated with SDXL. The comparison may not be fair. The visual comparison would be more appropriate if the same SD model is used for both methods, as demonstrated in Table 4's first two rows.

**Questions:**

1. Methods like chatGPT's new image generation model and Gemini's nano banana have shown impressive re-lighting performance recently. I wonder if the authors have tried to compare ReSWD's color matching results with these methods?

2. Lines 420-421: why a SDS-style gradient stopping "does not have any influence on the guidance"? Could the authors give expression of the gradient, similar to equation (3) in "DREAMFUSION: TEXT-TO-3D USING 2D DIFFUSION"?

3. See 2 and 3 in the Weaknesses section.

---

> ### Author Response · Authors · 2025-11-18
>
> We thank the reviewer for the suggestion. We introduced the following changes in the paper:
>
> ## Weakness (1):
>
> We consider the application implementation in itself as a contribution. It clearly outperforms the baselines in either variant (Base SWD or ours). We find small color shifts noticeable on good displays and they are highly important for professional work.
>
> ## Weakness (2):
>
> We apologize for the confusion. We indeed selected the largest K and fixed it in the revision.
>
> ## Weakness (3/4):
>
> We have added clarifications in Algorithm 1. This should now be clarified. We will also release the source code for future reference.
>
> ## Weakness 5:
>
> Thanks for the spot. We fixed it in the latest version.
>
> ## Weakness 6:
>
> Our contribution directly enables running on SD3.5, which was a limitation of the previous method. We showed the influence of each component in Table 4.
>
> | Method                         | CLIP-IQA ↑ | CLIP-T ↑ | Mean-W₂ [10²] ↓ | Time per generation [s] ↓ |
> |-------------------------------|------------|-----------|------------------|----------------------------|
> | Lobashev et al. (2025)        | 0.671      | 14.861    | 1.941            | 124                        |
> | Ours – Adam & GradStop @ SDXL | 0.691      | 14.848    | 1.832            | 126                        |
> | Ours – Adam @ SDXL            | 0.681      | 14.771    | 1.846            | 34                         |
> | Ours @ SDXL                   | 0.696      | 14.870    | 1.213            | 34                         |
> | Ours @ SD3.5-medium           | 0.786      | 14.783    | 0.815            | 32                         |
> | Ours @ SD3.5-turbo            | 0.800      | 14.882    | 0.675            | 4                          |
> | Ours @ SD3.5-large            | 0.793      | 14.817    | 0.550            | 69                         |
>
> ## Question 1:
>
> While these models are extremely powerful, they would regenerate the image. Besides unwanted hallucinations, the models are only capable of producing 8-bit image outputs. Most professional media works in higher bit-depth and wider gamuts. Our solution is just matching distributions and optimizing a CDL format which is agnostic to bit-depth and gamuts as well. Additionally, our solution also applies to videos as it is targeted at an industry standard for this format. Achieving perfect temporal consistency on videos is currently an unsolved problem for all video models.
>
> ## Question 2:
>
> We clarified this part in the revised submission. The influence is marginal, similar to Dreamfusion. The added benefits of faster inference and fewer memory requirements easily outweigh the slight performance degradation. Especially as it enables us to use a larger base model.

---

### Official Review · Reviewer_Mt3n · 2025-11-01

**Soundness:** 2
**Presentation:** 3
**Contribution:** 3
**Rating:** 4
**Confidence:** 2

**Summary:**

The paper proposes ReSWD, a Sliced Wasserstein Distance estimator that reduces gradient variance in optimization by reusing informative projection directions via Weighted Reservoir Sampling, inspired by ReSTIR. At each step, the method evaluates a pool of K (reservoir) + M (new) directions, keeps K survivors using WRS keys and computes a self-normalized estimate. A time-decay factor and ESS-based resets adapt to non-stationarity and prevent weight collapse. Empirically, ReSWD improves optimization stability and final quality across synthetic distribution matching, color correction, and diffusion color guidance, with modest runtime overhead. The idea is simple and broadly applicable. However, theoretical guarantees (especially unbiasedness under multi-item WRS, time decay, and self-normalized weighting) are insufficiently justified.

**Strengths:**

(i) The paper offers a simple, practical variance-reduction mechanism for SWD that is easy to integrate in differentiable pipelines

(ii) The paper shows consistent empirical gains over standard SWD, QMC, and control variates on synthetic and real tasks

(iii) The paper has a good selection of experiments for color correction with a Color Decision List and diffusion guidance

**Weaknesses:**

(i) Limited analysis of gradient variance reduction (no quantitative gradient variance or convergence-rate studies)

(ii) Although SDXL variants are reported, fairness could be clearer (model was switched to SD3.5 and the optimizer was changed)

(iii) Missing baseline against Max-SW, which the authors claim is closely related (see related work section)

(iv) Evaluation largely in low-dimensional color spaces (d=3) and lacks tests on higher-dimensional feature embeddings

(v) Some design choices lack sensitivity analysis beyond fresh-candidate ablation

(vi) The paper could benefit from more detailed derivations

**Questions:**

In addition to the weaknesses outlined in points (i-vi), I present the following questions for the authors to address:

(1) Please provide a formal derivation of the estimator: What are the exact inclusion probabilities under WRS for K>1, and how do they justify the weighting scheme in equation (7)?

(2) How do time-decay and ESS resets affect consistency? Are there any theoretical bounds?

(3) The paper argues about the stability of gradients. Can you report gradient variance (or signal-to-noise) over training steps vs. SWD/QMC?

(4) How does ReSWD perform with higher-dimensional features?

(5) Could you include Max-SW as a baseline, and clarify trade-offs vs. your approach?

(6) In diffusion guidance, can you isolate the contribution of ReSWD from other changes (optimizer, gradient stop) on the same base model with identical settings?

---

> ### Author Response · Authors · 2025-11-18
>
> We thank the reviewer for the suggestion. We introduced the following changes in the paper:
>
> ## Weakness (i) | Question (3) | Question (4):
>
> We have added the influence on gradients to Table 1. As seen there, our methods provide a drastically improved SNR compared to SWD, but it is weaker than the one from Max-SW. However, the optimization result from Max-SW is not as good as ours, and the optimization takes significantly longer. Intuitively, this comes from replacing the mean with a max estimator and only backpropagating through a single kernel. The added variance from visiting more features seems to be helpful in these settings. Our method strikes a balance between SWD and Max-SW by learning a reservoir of multiple good directions with minimal overhead.
>
> | Method                          | Mean-W₁ [10³] ↓ | Grad SNR ↑ | Time per step [ms] ↓ |
> |---------------------------------|------------------|-------------|------------------------|
> | SWD                             | 0.733            | 0.215       | 1.03                   |
> | Nguyen et al. (2024) (LCV)      | 0.735            | 0.218       | 1.81                   |
> | Nguyen et al. (2024) (UCV)      | 0.726            | 0.200       | 2.19                   |
> | Deshpande et al. (2019) (Max-SW)| 9.740            | 0.286       | 7.21                   |
> | Nguyen et al. (2024) (QMC)      | 0.670            | 0.209       | 1.51                   |
> | **Ours**                        | 0.622            | 0.278       | 1.92                   |
> | **Ours + QMC**                  | 0.610            | 0.274       | 2.10
>
> ## Weakness (ii) | Question 6:
>
> We added several additional ablations to Table 4, which showcase that our method is a larger contributor to the improvement compared to just adding Adam.
>
> | Method                         | CLIP-IQA ↑ | CLIP-T ↑ | Mean-W₂ [10²] ↓ | Time per generation [s] ↓ |
> |-------------------------------|------------|-----------|------------------|----------------------------|
> | Lobashev et al. (2025)        | 0.671      | 14.861    | 1.941            | 124                        |
> | Ours – Adam & GradStop @ SDXL | 0.691      | 14.848    | 1.832            | 126                        |
> | Ours – Adam @ SDXL            | 0.681      | 14.771    | 1.846            | 34                         |
> | Ours @ SDXL                   | 0.696      | 14.870    | 1.213            | 34                         |
> | Ours @ SD3.5-medium           | 0.786      | 14.783    | 0.815            | 32                         |
> | Ours @ SD3.5-turbo            | 0.800      | 14.882    | 0.675            | 4                          |
> | Ours @ SD3.5-large            | 0.793      | 14.817    | 0.550            | 69                         |
>
> ## Weakness (iv) | Question 4:
>
> We added higher dimensional evaluations; however, our method degrades at higher dimensionality. This is in line with our findings on extending the method to convolutions in the limitation section. There exist several three-dimensional problems where our method achieves consistent improvements over baselines.
>
> | Method                     | 3      | 6      | 12     | 24     | 32     | 48     | 64     |
> |----------------------------|--------|--------|--------|--------|--------|--------|--------|
> | SWD                        | 0.733  | 0.945  | 1.190  | 3.386  | 4.863  | 6.503  | 7.922  |
> | Nguyen et al. (2024) (LCV) | 0.735  | 0.963  | 2.203  | 4.186  | 5.271  | 7.152  | 8.826  |
> | Nguyen et al. (2024) (UCV) | 0.726  | 0.941  | 1.935  | 3.815  | 4.919  | 6.573  | 8.009  |
> | Nguyen et al. (2024) (QMC) | 0.670  | 0.736  | 1.166  | 3.904  | 4.899  | 6.509  | 7.854  |
> | **Ours**                   | 0.622  | 0.939  | 9.768  | 15.350 | 17.114 | 19.578 | 21.446 |
> | **Ours + QMC**             | 0.615  | 0.823  | 4.532  | 8.953  | 14.389 | 18.375 | 20.862 |
>
> ## Weakness (v) | Question 2:
>
> For the other parameters, we did not find any specific sensitivities. The number of directions has a similar effect to all SWD papers. Increasing the directions improves the results but becomes expensive. Too few directions, and the matching performance degrades. Similar to previous papers, we found 64 a good compromise. A too-low ESS alpha triggers the reset often, which degrades the method to SWD with M fewer filters. We found 0.5 a threshold which triggers in rare cases and prevents a collapse.
>
> ## Question 1:
>
> We defined the inclusion probability based on the contribution to the loss as seen in Section 3.2. Step 2. The weighting scheme in Eq. 7 is mainly used to keep unbiasedness, as we otherwise introduce overcontribution from the strongest influence, breaking unbiasedness.

---

### Author Response · Authors · 2025-11-18

We like to thank all reviewers for their feedback. We address each concern in a direct comment. The reviewers found that the key idea of “keeping `good’ directions for SWD gradient descent seems intuitive and useful” (3dAY, a2rM) with a2rM specifically highlighting the “Novel integration of rendering and optimal transport”. They also note that we “demonstrated the advantages of ReSWD on both real-world applications and synthetic distribution matching problems” (d1cU, Mt3n, Kxjt, a2rM). The method achieves “consistent empirical gains over standard SWD, QMC, and control variates” (Mt3n, d1cU, a2rM). They also remark that we provide a “good selection of experiments” (Mt3n, Kxjt, 3dAY, a2rM) and Kxjt specifically highlights the benefits of “Color Decision List, which greatly improves controllability for subsequent manual edits”. Furthermore, the paper is “well-written and the key challenge is clear” (3dAY, Kxjt) and 3dAY specifically emphasizes the explanatory video in the supplements.

We hope to alleviate most concerns in this rebuttal and that the reviewers will raise their scores accordingly. We have already incorporated the changes into the next revision of our manuscript.

---

### Note · Program_Chairs · 2026-01-17
**Submission Desk Rejected by Program Chairs**

The following references in this submission do not refer to real documents and/or have major errors in bibliographic information:

 Robin Rombach, Andreas Blattmann, and Björn Ommer. Text-to-image diffusion models in the wild. In ACM Transactions on Graphics (SIGGRAPH), 2022b.